# THE GAME OF HIDDEN RULES: A NEW CHALLENGE FOR MACHINE LEARNING

## ABSTRACT

Systematic examination of learning tasks remains an important but understudied area of machine learning (ML) research. To date, most ML research has focused on measuring performance on new tasks or surpassing state of the art performance on existing tasks. These efforts are vital but do not explain why some tasks are more difficult than others. Understanding how task characteristics affect difficulty is critical to formalizing ML's strengths and limitations; a rigorous assessment of which types of tasks are well-suited to a specific algorithm and, conversely, which algorithms are well-suited to a specific task would mark an important step forward for the field. To assist researchers in this effort, we introduce a novel learning environment designed to study how task characteristics affect measured difficulty for the learner. This tool frames learning tasks as a "board-clearing game," which we call the Game of Hidden Rules (GOHR). In each instance of the game, the researcher encodes a specific rule, unknown to the learner, that determines which moves are allowed at each state of the game. The learner must infer the rule through play. We detail the game's expressive rule syntax and show how it gives researchers granular control over learning tasks. We present example rules, an example ML algorithm, and methods to assess algorithm performance. Separately, we provide additional benchmark rules, a public leaderboard for performance on these rules, and documentation for installing and using the GOHR environment.

## 1 INTRODUCTION

Learning computational representations of rules has been one of the main objectives of the field of machine learning (ML) since its inception. In contrast to pattern recognition and classification (the other main domains of ML), rule learning is concerned with identifying a policy or computational representation of the hidden process by which data has been generated. These sorts of learning tasks have been common in applications of ML to real world settings such as biological research (Khatib et al., 2011), imitation learning (Hussein et al., 2018), and game play (Mnih et al., 2015; Silver et al., 2018). Since this process involves sequential experimentation with the system, much of the recent work exploring rule learning has focused on using reinforcement learning (RL) for learning rules as optimal policies of Markov decision processes.

An important question is whether some characteristics make particular rules easier or harder to learn by a specific algorithm (or in general). To date, this has been a difficult question to answer, since many rules of interest in the real world are multifaceted and not well characterized. For instance, while there are effective RL algorithms that can play backgammon, chess, and go, these games differ in significant ways and it is not clear how much each structural variation contributes to differences in overall difficulty for the learner. In order to investigate these questions, new ways of generating rules and data must be devised that allow for researchers to examine these characteristics in a controlled environment.

In this paper, we propose a new data environment called the Game of Hidden Rules (GOHR), which aims to help researchers in this endeavor. The main component of the environment is a game played in a $6 \times 6$ board with game pieces of different shapes and colors. The task of the learner is to clear the board in each round by moving the game pieces to "buckets" at the corners of the board according to a hidden rule, known to the researcher but not to the learner. Our environment allows researchers to express a hidden rule using a rich syntax that can map to many current tasks of interest

in both the classification and RL settings. A key advantage of our environment is that researchers can control each aspect of the hidden rules and test them at a granular level, allowing for experiments that determine exactly which characteristics make some learning tasks harder than others and which algorithms are better at learning specific types of rules.

The rest of the paper proceeds as follows. We first describe how our environment relates to other data-generating environments from the literature in Section 2. In Section 3, we describe the GOHR and its rule syntax, explaining how the environment can be used to investigate the effects of rule structure. We introduce our ML competition and refer readers to benchmark rules, instructions, and documentation available at our public site in Section 4. In Section 5, we present example rules and analysis for an example algorithm. Finally, we conclude with some discussion on the implications of our results and on other questions that can be answered by the GOHR environment in Section 6.

## 2 LITERATURE REVIEW

Games have historically served as rich benchmark environments for RL, with novel challenges in each game environment spurring progress for the field as a whole. RL has tackled increasingly complex classical board games, such as backgammon, chess, shogi, and go (Tesauro, 1994; Campbell et al., 2002; Silver et al., 2016; 2017; 2018), eventually surpassing the performance of human experts in each. Of late, video-game environments have also become drivers of progress in RL. Beginning with human-level performance in Atari 2600 games (Mnih et al., 2015; Badia et al., 2020), machine players have become competitive with humans in a variety of environments, including real-time-strategy and multiplayer games such as Quake III, StarCraft II, and Dota 2 (Jaderberg et al., 2019; OpenAI et al., 2019; Vinyals et al., 2017; 2019). Instrumental in this progress has been the growing space of benchmark environments and supporting tools, such as the Arcade Learning Environment (Bellemare et al., 2013), General Video Game AI (Perez-Liebana et al., 2016), OpenAI Gym (Brockman et al., 2016), Gym Retro (Nichol et al., 2018), Obstacle Tower (Juliani et al., 2019), Procgen (Cobbe et al., 2020), and NetHack environments (Küttler et al., 2020; Samvelyan et al., 2021). Taken together, these represent an impressive range of new tasks for RL agents, bridging many gaps and challenges in achieving aspects of artificial intelligence.

Benchmarks such as classic board and video games offer human-relatable assessments of ML but are not readily configurable. This makes it challenging to use them for the systematic study of task characteristics and their impact on learning. Recent literature has proposed variations on these environments, such as alternative forms of chess (Tomašev et al., 2022). While these environments provide additional insights, they are variations of already complex systems and do not provide the granularity needed to study the impact of task characteristics on learning. We propose the use of the GOHR environment to study these questions since it is constructed specifically for configurability.

GOHR distinguishes itself as a useful environment in four important ways. First, each hidden rule represents a deterministic mapping between the game's pieces and the four available buckets, allowing for clear distinctions between the characteristics of each learning task. Second, the game's rule syntax introduces a vast space of hidden rules for study, ranging from trivial to complex. Third, the rule syntax allows for fine variations in task definition, enabling experiments that study controlled differences in learning tasks. Fourth, encoding different hidden rules does not affect the learning environment. In contrast to existing benchmarking tools, this decouples the study of different learning tasks from associated changes to the environment itself, making it possible to compare the effects of task characteristics under identical environmental conditions.

## 3 GAME OF HIDDEN RULES

In this section we describe the GOHR's structure, syntax, and the expressivity of the rule language. In each episode of our game, the player is presented with a game board containing game pieces, each drawn from a configurable set of shapes and colors. The player's objective is to remove game pieces from play by dragging and dropping them into buckets located at each corner of the game board. A hidden rule, unknown to the player, determines which pieces may be placed into which buckets at a given point in the game play. For instance, a rule might assign game pieces to specific buckets based on their shape or color. If the player makes a move permitted by the rule, the corresponding game piece is removed from play; otherwise, it remains in its original location. The episode concludes

once the player has fully satisfied the rule. Typically, this occurs when all game pieces have been cleared from play, but some rules may be fully satisfied even with some pieces remaining on the board. The scoring for GOHR rewards players for completing episodes in as few moves as possible, incentivizing players to quickly discern the hidden rule.

The GOHR is played on a board with 36 identical cells, arranged in a $6 \times 6$ grid. Each cell on the board is indexed with a label, 1-36, and has $x$ and $y$ coordinates 1-6. Each bucket is indexed with a label, 0-3, with $x, y$ coordinates in the set $\{(0,0), (0,7), (7,0), (7,7)\}$. A diagram of the game board, including all numeric labels, can be seen in Figure 1.

In each GOHR experiment, the game engine generates game pieces from user-defined sets of shapes and colors. Depending on the experimental goals, these sets can be of arbitrary size. If no specification is provided in the experimental setup, the game engine defaults to a set of 4 shapes (circles, triangles, squares, stars) and 4 colors (red, blue, black, yellow). The experiment designer may add shapes or colors to the experiment in the associated color and shape configuration files. A sample board, built using a set of 4 shapes and 4 colors, is shown in Figure 2.

| | Columns | | | | | | |
|---|---|---|---|---|---|---|---|
| (Bucket 0) | 1 | 2 | 3 | 4 | 5 | 6 | (Bucket 1) |
| Row 6 | 31 | 32 | 33 | 34 | 35 | 36 | |
| Row 5 | 25 | 26 | 27 | 28 | 29 | 30 | |
| Row 4 | 19 | 20 | 21 | 22 | 23 | 24 | |
| Row 3 | 13 | 14 | 15 | 16 | 17 | 18 | |
| Row 2 | 7 | 8 | 9 | 10 | 11 | 12 | |
| Row 1 | 1 | 2 | 3 | 4 | 5 | 6 | |
| (Bucket 3) | | | | | | | (Bucket 2) |

Figure 1: Game board diagram, with numeric position labels included.

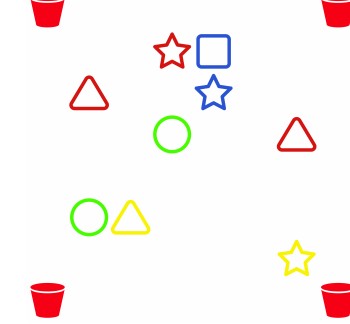

Figure 2: Game board, configured with 4 colors and 4 shapes, as it would be seen by a human player.

Boards in the GOHR can be defined in advance of play (to permit precise experimenter control), or can be generated randomly by the game engine according to a set of input parameters. This flexibility allows the experimenter to evaluate the impact of deterministic or randomized "curricula" (e.g., to determine whether seeing a particular set of boards in a specified order improves learning). When generating boards randomly, the experimenter must specify the minimum and maximum numbers of game pieces, colors, and shapes to appear on each new board; the game engine randomly selects values in the range [minimum, maximum] for each quantity and generates boards accordingly. Additional documentation can be found at our public site, as noted in Section 4.

### 3.1 RULE DEFINITION

A rule playable within GOHR is defined in a standalone file. Each rule is constructed from one or more **rule lines**, each of which is built from one or more **atoms**. For instance, a two-line rule with five atoms might look like:

$$\text{(atom 1) (atom 2) (atom 3)}$$
$$\text{(atom 4) (atom 5)}$$

Only one rule line is **active** at a time; this active line determines the current **rule state** (how game pieces may be placed into buckets for the player's next move). In the example above, the rule state is formed by the contents of either atoms 1, 2, and 3 or atoms 4 and 5, depending on which line is active. Later sections discuss the mechanisms by which the active rule line can change during play.

Each atom maps a set of game pieces to a set of permitted buckets and is defined as follows:

(count, shapes, colors, positions, buckets)

The "count" parameter defines the number of successful moves that the atom permits (also referred to as its "metering"). Any game pieces matching the "shapes," "colors," and "positions" specified in the atom are accepted in any element of the set of "buckets." Multiple values can be listed for each non-count field and are grouped in square brackets. For example, the rule that "stars and triangles go in bucket 0, squares in the bottom 3 rows go in bucket 1, squares in the top 3 rows go in bucket 2, and circles go in buckets 2 or 3" would be constructed with the following four atoms on a single line:

(*,[star, triangle],*,*,0) (*,square,*,[R1,R2,R3],1) (*,square,*,[R4,R5,R6],2) (*,circle,*,*,[2,3])

The $*$ character is a wildcard. For the count field, $*$ means the atom is not metered; for shape, color, or position, it means any value for that feature is permissible. In this example, atoms apply to any indicated shape regardless of their color. Since the atoms are not metered, the objects may be removed any time that rule line is active. For brevity, in the above, we use shorthand to refer to each row of the game board (R#); please refer to the detailed manual noted in Section 4 for additional notes on position-related syntax.

## 3.2 RULE EXPRESSIVITY

The above syntax permits a vast range of rules within the GOHR. **Stationary** rules, such as the previous example, are those where the permitted bucket set for each game piece never changes during play. Thus, stationary rules permit the experimenter to study the practical difficulty associated with learning feature-based patterns–i.e., rules that map game pieces to buckets based on their color, shape, position, or a combination thereof. Though stationary rules have no dependence on play history, these rules can still be made difficult, such as by designating a specific bucket for every possible combination of a game piece's features. Importantly, since the set of colors and shapes used is left to the experimenter, such rules can become arbitrarily complex.

**Non-stationary** rules are those in which the permitted bucket set for game pieces changes during play. Non-stationary rules can involve following a defined sequence, such as clearing a triangle, a square, a circle, and then a star in that order (repeating the sequence as needed). It is also possible to create non-stationary rules with an implicit priority between objects, such as clearing all available red pieces before all available blue pieces, or clearing pieces on the board from top to bottom. Methods for implementing non-stationary rules are described below. Whether stationary or non-stationary, we emphasize that all rules within GOHR are deterministic.

As mentioned, each atom contains a "count" field dictating the number of corresponding correct moves that the atom permits. When a player makes a move satisfying one (or more) of the atoms on the active rule line, the counts associated with the satisfied atoms are decremented by one. An atom with a count of 0 is considered exhausted and no longer accepts moves. At each move, the game engine evaluates the active rule line and the current board; if there are no valid moves available to the player, the engine makes the next rule line active and resets all counts in the new line. If there are no lines below the current rule line, the first rule line is made active again. This functionality can be used to encode sequences into the hidden rules. For example, the following two rules require the player to alternate between placing game pieces in the top (0,1) and bottom (2,3) buckets, with subtly different mechanics.

| Strict Alternation | Ambiguous Alternation |
| --- | --- |
| (1, *, *, *, [0,1]) | (1, *, *, *, [0,1]) (1, *, *, *, [2,3]) |
| (1, *, *, *, [2,3]) | |

In the strict case, the player's first move is allowed into only the top buckets and every correct move exhausts the current rule line. As a result, the active rule line will alternate with each correct move until the board is cleared. In the ambiguous case, a similar alternation occurs, but the order depends on the player's first correct move. Since both atoms are on the same rule line, the player's first move may go in any of the four buckets. After one successful move, the player may only make a move satisfying the remaining, non-exhausted atom. After two successful moves, all atoms in the rule line are exhausted and both are reset. This process repeats until the board is cleared.

GOHR also permits the experimenter to attach a count to each rule line. When the line is active, this count decrements each time any atom on the line is satisfied. For instance, a rule that alternates between uniquely assigning shapes and colors to buckets would look like:

$$1 (*, star, *, *, 0) (*, square, *, *, 1) (*, circle, *, *, 2) (*, triangle, *, *, 3)$$
$$1 (*, *, black, *, 0) (*, *, yellow, *, 1) (*, *, blue, *, 2) (*, *, red, *, 3)$$

If the rule-line count is exhausted, the game engine makes the next line active, even if there are non-exhausted atoms on that line. For this example, the active rule line will alternate after each successful move, regardless of which atom in the active line is satisfied. If no count is provided for a given rule line, the game engine assumes that the rule line is not metered and can be used until all atoms on that line are exhausted or no valid move exists for the game pieces currently on the board.

The GOHR rule syntax allows the experimenter to write expressions in an atom's bucket field that define sequences based on previous successful moves. The game engine stores values for the bucket that most recently accepted any game piece ($p$) as well as the bucket that most recently accepted an item of each color ($pc$) and shape ($ps$). A simple rule expressible using these values is "objects must be placed in buckets in a clockwise pattern, beginning with any bucket":

$$(1, *, *, *, [0,1,2,3])$$
$$(*, *, *, *, p+1)$$

The expressions used in the buckets field are evaluated modulo 4 to ensure that the resultant expression gives values in the range 0-3. The game engine also supports the terms "nearby" and "remotest" as bucket definitions, which allow the experimenter to require that game pieces be put into either the closest or furthest bucket, respectively, evaluated by Euclidean distance.

The arrangement of atoms allows the experimenter to encode priority between component tasks within a rule. For instance, the rule that "all red pieces go into bucket 1, *then* all blue pieces go into bucket 2" would be expressed as follows:

$$(*, *, red, *, 1)$$
$$(*, *, blue, *, 2)$$

Since both the first rule line and associated atom are not metered, they can never become exhausted, even if there are no more red game pieces left on the board. However, as noted above, the engine evaluates if there are any valid moves available to the player given the current rule line; if there are no valid moves available, the engine shifts control to the next line. In this example, once the player has cleared all red game pieces from the board, the engine will make the second rule line active.

## 4    DOCUMENTATION AND COMPETITION

GOHR is designed to be a flexible benchmarking environment for testing various rule learning related hypotheses. Beyond using GOHR for the study of task characteristics and their impact on difficulty, we hope that researchers will share performance results on benchmark rules with us so that we can disseminate community-wide benchmark performance. To this effect, we plan to host a public leader-board through June 2023 that will present results for benchmark rules, updated on a weekly basis. The public site houses documentation for downloading and setting up the captive game server (CGS), resources for configuring and running experiments, benchmark rules supplementing those presented in Section 5, as well as the public leader-board.

## 5    EXAMPLE ANALYSIS

To illustrate how GOHR can be used for comparing rule difficulty, we provide a few example rules and test them using one specific ML algorithm (MLA). We use these example results to show possible performance metrics, statistical tests, and data presentation methods. Though we briefly discuss structural differences in the proposed example rules, the rules were chosen primarily to illustrate a range of possibilities within the game; we expect that researchers will design targeted sets of rules to assess the impact of particular task characteristics on learning difficulty. Likewise, researchers may

perform similar analyses using different learning algorithms to compare performance across common rules. Here we use a reinforcement-learning algorithm, but batch methods applied to, for example, data generated by random moves, could also be studied with GOHR, and may reveal unexpected relations regarding the difficulty of learning particular rules.

## 5.1 EXAMPLE RULES

We discuss the following example rules, played using the default four shapes and colors. Boards were randomly generated with nine game pieces each, including at least one piece of every shape and color. We note that the first rule is stationary, while the latter three rules are non-stationary.

1. **Color Match**: Each of the four default colors goes in a specific bucket, in any order.

   (*, *, black, *, 0) (*, *, yellow, *, 1) (*, *, blue, *, 2) (*, *, red, *, 3)

2. **Clockwise**: The first game piece can go in any bucket, but each subsequent piece must go in the next clockwise bucket

   (1, *, *, *, [0,1,2,3])

   (*, *, *, *, (p + 1))

3. **B23 then B01**: Game pieces must be alternatively placed in the bottom and top buckets

   (1, *, *, *, [2,3])

   (1, *, *, *, [0,1])

4. **B3 then B1**: Game pieces must be alternatively placed in buckets 3 and 1

   (1, *, *, *, 3)

   (1, *, *, *, 1)

## 5.2 EXAMPLE MACHINE LEARNING ALGORITHM

We model the GOHR as an episodic Markov decision process (MDP) characterized by the tuple, $\mathcal{M} = (\mathcal{S}, \mathcal{A}, R, P, \mu, H)$. $\mathcal{A}$ is the action space, where each action puts the object in row $r$ and column $c$ in bucket $b$: $\mathcal{A} = \{(r, c, b) : r \in [6], c \in [6], b \in \{(0, 0), (0, 7), (7, 0), (7, 7)\}\}$. Before introducing the state space, we first formally describe the board. In a board $B$, the cell at row $r$, column $c$ is characterized by color and shape: $B[r, c].color \in \{$red, black, blue, yellow, $\emptyset\}$, $B[r, c].shape \in \{$star, square, triangle, circle, $\emptyset\}$, where we define the shape and color of an empty cell to be $\emptyset$. $\mathcal{S}$ is the state space, where each state $S_t = (B_0, A_0, B_1, A_1, \cdots, B_{t-1}, A_{t-1}, B_t)$ is a sequence including all the historical boards $B_i$'s and actions $A_i$'s up to time step $t$. We define $B_{-1}$ to be an empty board before the board generation process. $P$ is the deterministic transition probability matrix specified by the hidden rule. Importantly, when action $A_t$ is not accepted, resulting in no change to the board, we still insert the board $B_{t+1}$ and action $A_t$ to $S_{t+1}$. In that case, $B_{t+1} = B_t$. $R$ is the deterministic reward function. If an action is accepted or the board is already cleared, the reward is set to be 0; otherwise, the reward is $-1$. Specifically: $R(S_t, A_t) = 0$ if $B_{t+1} \neq B_t$ or $B_t = B_{-1}$; $R(S_t, A_t) = -1$, otherwise $\mu$ is the initial state distribution. Since the initial state $S_0 = (B_0)$, which only includes the initial board, $\mu$ is determined by the board generating process in Section 3. $H$ is the time horizon that characterizes the number of time steps in each episode. The MDP evolves according to Section 3. We define the value function as the sum of future rewards by taking action according to policy $\pi$, i.e. $V^\pi(S_t) = \mathbb{E}\left[\sum_{k=t}^{H-1} R(S_k, A_k)\right]$, where the expectation is over the potential randomness of the policy. Similarly, we define the state-action value function by the sum of future rewards by taking action $A_t$ at state $S_t$ and following policy $\pi$ after that, i.e. $Q^\pi(S_t, A_t) = R(S_t, A_t) + V^\pi(S_{t+1})$. Because the step information has already been encoded in the state, we do not need to use step $h$ as the subscript for the value function and state-action value function. The goal of the learner is to find a policy $\pi$ to maximize the expected cumulative reward $\mathbb{E}_{S_0 \sim \mu} [V^\pi(S_0)]$.

Even though the MDP $\mathcal{M}$ has a finite state space $\mathcal{S}$, the size of $\mathcal{S}$ scales exponentially in $H$, which makes it impractical to learn a proper policy directly. Instead, we handcraft a feature mapping $\phi(\cdot, \cdot)$ to extract only the essential information for some particular hidden rules. The feature function maps each state-action pair $(S_t, A)$ to a 3720-dimensional Boolean vector $\phi(S_t, A)$. Recall that each action $A$ chooses a cell $(r, c)$ and puts the object in this cell in bucket $b$. $\phi(S_t, A)$ encodes the following

information to Boolean variables: the shape and color information of the cell chosen by $A$, the target bucket, and the shape, color, and bucket information of the last accepted action. Products of some of the Boolean variables are also included in $\phi(S_t, A)$. Interested readers are referred to Appendix A.1.1 for additional details on the feature representation.

We use a variant of DQN (Mnih et al., 2013), a model-free algorithm, as our learning algorithm. Our MLA models the state-action value function $Q$ as a linear function in the feature vector $\phi$: $Q(s, a; \theta) = \theta^\top \phi(s, a)$ and tries to estimate the optimal $Q^*$ using stochastic gradient updates on moving loss $L$:

$$L(\theta) = \mathbb{E}\left[(y - Q(s, a; \theta))^2\right],$$

where $y = r + \max_{a'} Q(s', a'; \theta^{\text{target}})$ is the target $Q$ estimate obtained for the sample $(s, a, r, s')$. We distinguish the running parameter iterate $\theta$ and the target parameter $\theta^{\text{target}}$. Unlike the DQN in Mnih et al. (2013), $\theta^{\text{target}}$ is updated less frequently to stabilize the algorithm.

Additionally, the algorithm maintains a buffer memory of size 1000 for past experiences and samples a batch of 128 tuples to calculate gradients at each time step. We run the MLA with an $\epsilon$-greedy behavior policy. The $\epsilon$ starts at 0.9 and terminates at 0.001. $\epsilon$ decays exponentially with step-size 200. More precisely, $\epsilon = 0.001 + (0.9 - 0.001) \exp(-\text{number of moves}/200)$. For each rule we run 100 trials, with each trial including 200 episodes of length $H = 100$. The full learning algorithm is described in Appendix A.1.2 and relevant hardware information is described in Appendix A.1.3.

### 5.3 METRICS AND EXAMPLE RESULTS

In each independent learning run for a given rule, we train our MLA from scratch by playing a set number of randomly generated episodes of that rule; each learning run uses different random seeds for the MLA and the game engine-board generator. We measure a rule's difficulty using the cumulated number of errors made across the episodes of each independent learning run. As an MLA plays and learns from its experiences, its error rate decreases and the cumulated error count for that learning run levels off. This indicates that the agent has found a policy that can play the rule essentially error-free. Rules that are more difficult to learn result in more cumulated errors before the agent arrives at an error-free policy. By recording many learning runs, we obtain a distribution of cumulated error values that can be used to perform statistical comparisons between rules. For our example rules and MLA, 200 episodes were sufficient for the cumulated error curves to flatten; we refer to the cumulated error after 200 episodes as the Terminal Cumulated Error (TCE) for that learning run. For other rules or algorithms, longer learning runs might be necessary. Figure 3 shows cumulated error curves of our example MLA for 100 separate learning runs of the Color Match rule (each consisting of 200 episodes). Such figures give insight into the shape of the TCE distribution for that rule and MLA. In addition, these data can be used to generate median learning curves that show the median cumulated error in each episode and associated 95% confidence intervals across all runs, as seen in Figure 4. Median learning curves visually rank rule difficulty, with higher curves indicating greater difficulty. Box-and-whisker plots for the distributions of TCE for each example rule are shown in Figure 5, with higher distributions again indicating greater difficulty.

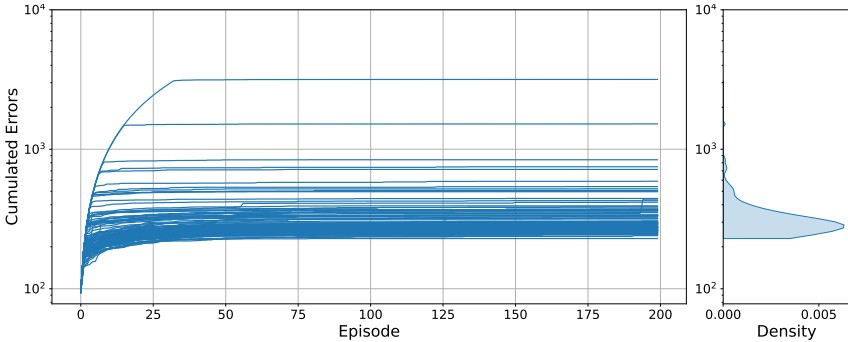

Figure 3: Cumulated error curves for 100 MLA learning runs of the Color Match rule. The y-axis is logarithmic to capture the distribution's skewness. The marginal plot to the right displays a kernel density estimate for the TCE for this rule and MLA.

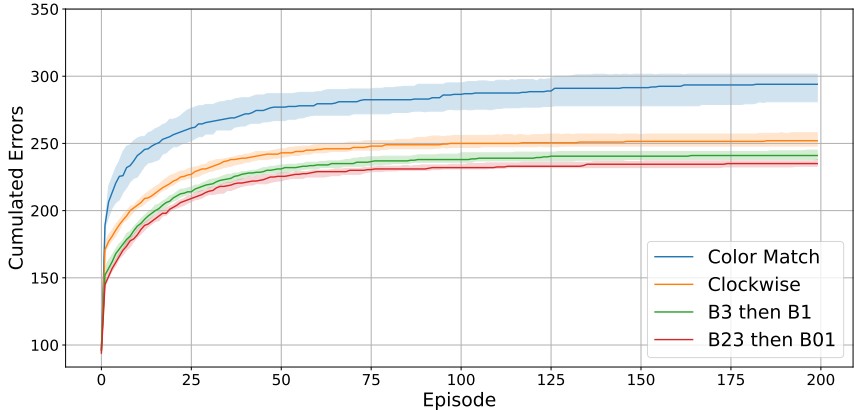

Figure 4: Median cumulated errors at each episode across 100 MLA learning runs for each rule. The shaded regions denote 95% confidence intervals for the medians, calculated with 50,000 bootstraps.

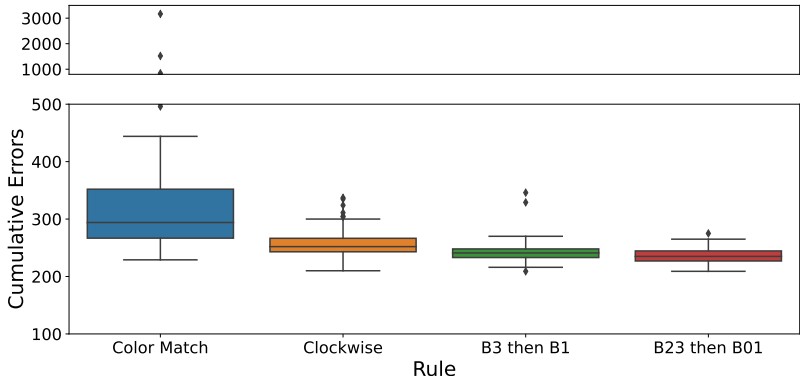

Figure 5: Box-and-whisker plots of TCE for each learning run for all example rules. The chart has a broken y-axis with a scale change to capture the most extreme outliers. The colored area is the interquartile range (IQR). Whiskers denote data within 1.5 IQRs of the upper and lower quartiles.

Although the heavily-skewed TCE distribution seen for Color Match was not seen for all example rules, it highlights the need for non-parametric tests when comparing the difficulty of two or more rules. To perform pairwise comparison of rule difficulty, for instance between two specific rules, $A$ and $B$, we recommend using the Mann-Whitney-Wilcoxon U-Test (Nachar, 2008) with TCE as the point-metric. The test compares every TCE value associated with rule $A$ against every TCE value associated with rule $B$; the associated test statistic sums the number of times that values from $A$ exceed values from $B$, with ties counting for 0.5. The difficulty ranking seen in Figure 4 was confirmed by pairwise one-sided U-Tests; the largest $p$ value across all pairwise tests was less than 0.002.

Although this analysis is intended only to illustrate the process for measuring algorithm performance, we note a few interesting structural differences in our example rules which could merit further study. We believe a framework for analyzing task characteristics in the GOHR should consider the number of underlying logical mappings in a rule, the arrangement of these mappings across features or sequential mechanics, and the cardinality of these mappings (e.g. 1:1, 1:n, etc.). Under this framework, Color Match requires that the player learn four 1:1 mappings between colors and buckets. Clockwise requires that the player learn four 1:1 mappings between the prior bucket in the clockwise sequence and the next bucket in the sequence. Like Clockwise, B3 then B1 considers 1:1 mappings based on the last correct move, but the player must learn only two such mappings rather than four. B23 then B01 requires the player to learn two sequential 1:1 mappings, similar to B3 then B1, but the player can learn any one of four pairs of mappings as part of an error-free policy. Despite their structural differences, we note that Color Match, Clockwise, and B3 then B1 each give the player

the same probability of making correct moves at random ($1/4$); we thus refer to these rules as being equally permissive. By this definition, B23 then B01 is the most permissive rule in this set of example rules, as it has a corresponding probability of $1/2$.

Recalling the difficulty ordering shown in Figure 4, confirmed by pairwise U-tests, the rules with four underlying mappings (Color Match and Clockwise) were more difficult for our example MLA than those with only two (B3 then B1 and B23 then B01). B3 then B1 was found to be more difficult than the more permissive B23 then B01. Color Match and Clockwise require learning the same number of mappings, but Color Match was measured to be more difficult. Although Color Match is stationary and Clockwise is non-stationary, we believe this performance difference is primarily an artifact of our MLA's featurization method. These learning tasks are only meant to suggest interesting avenues for future research; we believe that the GOHR equips researchers to employ larger and more granular rule sets to pursue ideas similar to those raised here in order to rigorously identify how particular task characteristics affect difficulty for a given learning algorithm.

In addition to comparing the difficulty of two different rules for a given algorithm, the U-Test also supports comparison between different MLAs. The U-Test statistics can be used to form an "ease ratio," representing the ratio of learning runs where rule $A$ was found easier than rule $B$ by a given MLA. We propose aggregating ease ratios for many MLAs across sets of benchmark rules to create a tournament graph as described in Strang et al. (2022). The ranking method corrects for the fact that algorithms $R$ and $S$ may have solved a different subset of rules, leading to the possibility that $R$ encountered a larger fraction of "easy" rules (where rule difficulty can be assessed by the TCE of all algorithms that attempt a particular rule). Additional information regarding the methods to be used in ranking competition submissions can be found at the public site.

## 6    DISCUSSION

The GOHR tool provides a capability for studying the performance of MLAs in a novel and principled way. Using the expressive rule syntax, researchers can make precise changes to rules of interest to study how they affect algorithm performance. As suggested in Section 5, this approach can be used to compare the performance of different algorithms on any specific set of rules and to see the effect of rule characteristics on MLA performance. This is a useful complement to present competitions among ML algorithms on existing board games and video games that can look only at complex tasks without a fine level of granularity. The original proposers, Bier et al. (2019), envisioned a simpler version of what we call the GOHR as a tool for comparing human and machine learning; here we extend this notion to a probe of differing ML frameworks.

Each choice of ML parametrization, learning model, and optimization is expected to produce different results. With the GOHR, we believe it will be possible to explore how these choices interact with specific rule characteristics to result in differences in practical performance. With this goal in mind, we are sharing the complete suite of tools with all interested researchers. We hope that researchers using the GOHR tool for ML research will share their findings to help this inquiry.

We believe there are at least two interesting broad research questions that can be answered with the GOHR environment. First, can a given ML algorithm learn many substantially different rules? Algorithms capable of this kind of general learning within the GOHR environment may prove more capable in other domains. Second, when results for many algorithms are available, will it be possible to cluster rules into intrinsic difficulty classes based on those experimental findings? Such a classification may reveal conceptual task characteristics, extensible beyond GOHR, that characterize learning complexity more generally. We hope this platform will be helpful to ML researchers in answering such questions.

## 7    REPRODUCIBILITY STATEMENT

Our paper presents example results intended to show the expressivity of the rule syntax as well as example means of creating and analyzing experiments; these results are not intended to serve as a specific performance benchmark. Regardless, we provide the code used to generate these results in the supplementary materials, with corresponding instructions in the Appendix so that they can be reproduced by interested researchers.

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

# A APPENDIX

## A.1 DETAILS OF THE REINFORCEMENT LEARNING ALGORITHM

### A.1.1 FEATURE REPRESENTATION

The feature vector $\phi(S, A)$ is constructed with the following four different types of components:

1. The 12 current-step unary features $\phi_u(S, A)$, where $u$ can be a color, a shape, or a bucket. If $u$ is a color:

$$\phi_u(S_t, A) = \mathbb{1}\{B_t[A.r, A.c].\text{color} = u\}.$$

If $u$ is a shape:

$$\phi_u(S_t, A) = \mathbb{1}\{B_t[A.r, A.c].\text{shape} = u\}.$$

If $u$ is a bucket:

$$\phi_u(S_t, A) = \mathbb{1}\{A.b = u\}.$$

2. The 48 current-step binary features, a set of binary features that depend on the current board configuration $\phi_{u,v} : \mathcal{S} \times \mathcal{A} \to \{0, 1\}$ where $(u, v)$ can be a (color, shape), (color, bucket) or (shape, bucket) tuple. These features $\phi_{u,v}(S_t, A)$ are defined as follows:

| $u, v \in$ | $\phi_{u,v}(S_t, A)$ |
|---|---|
| color $\times$ shape | $\mathbb{1}\{B_t[A.r, A.c].\text{color} = u\} \wedge \mathbb{1}\{B_t[A.r, A.c].\text{shape} = v\}$ |
| color $\times$ bucket | $\mathbb{1}\{B_t[A.r, A.c].\text{color} = u\} \wedge \mathbb{1}\{A.b = v\}$ |
| shape $\times$ bucket | $\mathbb{1}\{B_t[A.r, A.c].\text{shape} = u\} \wedge \mathbb{1}\{A.b = v\}$ |

3. The 60 last-successful-step and current-step binary features, a set of binary features constructed from the board and action information at the last successful step and the current step. For a time step $t > 0$, let

$$\ell(t) = \max\{\tau \in \{-1, 0, \cdots, t-1\} \text{ s.t. } B_t \neq B_\tau\}$$

be the latest successful time step. Since $B_{-1}$ is an empty board, $B_{-1} \neq B_t, \forall t \geq 0$ until the game is over, so $\ell(t)$ is well defined.

The features are defined as follows: $\phi_{(u),v} : \mathcal{S} \times \mathcal{A} \to \{0, 1\}$, where $(u)$ is extracted from the board and action information at the last successful time step, and $v$ is extracted from the board and action information at the current time step. Note that $u$ can be the empty set $\emptyset$ when $\ell(t) = -1$.

| $(u), v \in$ | $\phi_{(u),v}(S_t, A)$ |
|---|---|
| $(\text{color}^\emptyset) \times \text{color}$ | $\mathbb{1}\{B_{\ell(t)}[A_{\ell(t)}.r, A_{\ell(t)}.c].\text{color} = u\} \wedge \mathbb{1}\{B_t[A.r, A.c].\text{color} = v\}$ |
| $(\text{shape}^\emptyset) \times \text{shape}$ | $\mathbb{1}\{B_{\ell(t)}[A_{\ell(t)}.r, A_{\ell(t)}.c].\text{shape} = u\} \wedge \mathbb{1}\{B_t[A.r, A.c].\text{shape} = v\}$ |
| $(\text{bucket}^\emptyset) \times \text{bucket}$ | $\mathbb{1}\{A_{\ell(t)}.b = u\} \wedge \mathbb{1}\{A.b = v\}$ |

where $x^\emptyset = x \cup \{\emptyset\}, \forall x \in \{\text{color}, \text{shape}, \text{bucket}\}$

4. The 3600 last-successful-step and current-step 4-ary features, a set of 4-ary features constructed from the board and action information at the last successful step and the current step.

$$\phi_{(u,v),y,z} : \mathcal{S} \times \mathcal{A} \to \{0, 1\}$$

where $(u, v)$ are extracted from the board and action information at the last successful time step, and $y, z$ are extracted from the current time step. These features, $\phi_{(u,v),y,z}(S_t, A)$, are defined as follows:

| $(u,v), y, z \in$ | $\phi_{(u,v),y,z}(S_t, A)$ |
|---|---|
| $(\text{shape}^\emptyset, \text{color}^\emptyset)$ $\times \text{shape}, \text{color}$ | $\mathbb{1}\{B_{\ell(t)}[A_{\ell(t)}.r, A_{\ell(t)}.c].\text{shape} = u\}$ $\wedge \mathbb{1}\{B_{\ell(t)}[A_{\ell(t)}.r, A_{\ell(t)}.c].\text{color} = v\}$ $\wedge \mathbb{1}\{B_t[A.r, A.c].\text{shape} = y\} \wedge \mathbb{1}\{B_t[A.r, A.c].\text{color} = z\}$ |
| $(\text{shape}^\emptyset, \text{color}^\emptyset)$ $\times \text{shape}, \text{bucket}$ | $\mathbb{1}\{B_{\ell(t)}[A_{\ell(t)}.r, A_{\ell(t)}.c].\text{shape} = u\}$ $\wedge \mathbb{1}\{B_{\ell(t)}[A_{\ell(t)}.r, A_{\ell(t)}.c].\text{color} = v\}$ $\wedge \mathbb{1}\{B_t[A.r, A.c].\text{shape} = y\} \wedge \mathbb{1}\{A.b = z\}$ |
| $(\text{shape}^\emptyset, \text{color}^\emptyset)$ $\times \text{color}, \text{bucket}$ | $\mathbb{1}\{B_{\ell(t)}[A_{\ell(t)}.r, A_{\ell(t)}.c].\text{shape} = u\}$ $\wedge \mathbb{1}\{B_{\ell(t)}[A_{\ell(t)}.r, A_{\ell(t)}.c].\text{color} = v\}$ $\wedge \mathbb{1}\{B_t[A.r, A.c].\text{color} = v\} \wedge \mathbb{1}\{A.b = z\}$ |
| $(\text{shape}^\emptyset, \text{bucket}^\emptyset)$ $\times \text{shape}, \text{color}$ | $\mathbb{1}\{B_{\ell(t)}[A_{\ell(t)}.r, A_{\ell(t)}.c].\text{shape} = u\} \wedge \mathbb{1}\{A_{\ell(t)}.b = v\}$ $\wedge \mathbb{1}\{B_t[A.r, A.c].\text{shape} = y\} \wedge \mathbb{1}\{B_t[A.r, A.c].\text{color} = z\}$ |
| $(\text{shape}^\emptyset, \text{bucket}^\emptyset)$ $\times \text{shape}, \text{bucket}$ | $\mathbb{1}\{B_{\ell(t)}[A_{\ell(t)}.r, A_{\ell(t)}.c].\text{shape} = u\} \wedge \mathbb{1}\{A_{\ell(t)}.b = v\}$ $\wedge \mathbb{1}\{B_t[A.r, A.c].\text{shape} = y\} \wedge \mathbb{1}\{A.b = z\}$ |
| $(\text{shape}^\emptyset, \text{bucket}^\emptyset)$ $\times \text{color}, \text{bucket}$ | $\mathbb{1}\{B_{\ell(t)}[A_{\ell(t)}.r, A_{\ell(t)}.c].\text{shape} = u\} \wedge \mathbb{1}\{A_{\ell(t)}.b = v\}$ $\wedge \mathbb{1}\{B_t[A.r, A.c].\text{color} = y\} \wedge \mathbb{1}\{A.b = z\}$ |
| $(\text{color}^\emptyset, \text{bucket}^\emptyset)$ $\times \text{shape}, \text{color}$ | $\mathbb{1}\{B_{\ell(t)}[A_{\ell(t)}.r, A_{\ell(t)}.c].\text{color} = u\} \wedge \mathbb{1}\{A_{\ell(t)}.b = v\} \wedge$ $\mathbb{1}\{B_t[A.r, A.c].\text{shape} = y\} \wedge \mathbb{1}\{B_t[A.r, A.c].\text{color} = z\}$ |
| $(\text{color}^\emptyset, \text{bucket}^\emptyset)$ $\times \text{shape}, \text{bucket}$ | $\mathbb{1}\{B_{\ell(t)}[A_{\ell(t)}.r, A_{\ell(t)}.c].\text{color} = u\} \wedge \mathbb{1}\{A_{\ell(t)}.b = v\}$ $\wedge \mathbb{1}\{B_t[A.r, A.c].\text{shape} = y\} \wedge \mathbb{1}\{A.b = z\}$ |
| $(\text{color}^\emptyset, \text{bucket}^\emptyset)$ $\times \text{color}, \text{bucket}$ | $\mathbb{1}\{B_{\ell(t)}[A_{\ell(t)}.r, A_{\ell(t)}.c].\text{color} = u\} \wedge \mathbb{1}\{A_{\ell(t)}.b = v\}$ $\wedge \mathbb{1}\{B_t[A.r, A.c].\text{color} = y\} \wedge \mathbb{1}\{A.b = z\}$ |

For any $(S, A)$-tuple, the final feature vector $\phi(S, A)$ is created by stacking up all of the above features, which gives a feature function $\phi : \mathcal{S} \times \mathcal{A} \to \{0, 1\}^{3720}$.

### A.1.2  LEARNING ALGORITHM

---

**Algorithm 1** Linear Variant of Deep Q-learning(DQN) with Experience Replay

---

**Input** : target iteration $T$, episode count $M$, buffer size $N$, batch size $B$.

Initialize experience replay buffer memory $\mathcal{D}$ with capacity $N$.
Initialize the iterate parameters $\theta_1^{\text{target}} \leftarrow 0, \theta_{1,1} \leftarrow 0$.
**for** target iteration $\tau = 1, \cdots, T$ **do**
    **for** episode $e = 1, \cdots, M$ **do**
        Draw $s_1 \sim \mu$.
        **for** time step $t = 1, \cdots, H$ **do**
            Draw $a_t \sim \epsilon \cdot \text{Unif}(A) + (1 - \epsilon) \cdot \mathbb{1}_A\{a = \max_{a'} Q(s_t, a'; \theta_{e,t})\}$.
            Take action $a_t$ and observe reward $r_t$ and observation $x_{t+1}$.
            Set $s_{t+1} \leftarrow (s_t, a_t, x_{t+1})$ and enque $(s_t, a_t, r_t, s_{t+1})$ to $\mathcal{D}$.
            Uniformly sample a batch $\{(s_j, a_j, r_j, s'_j)\}|_{j=1}^B$ from $\mathcal{D}$ and
              set $\forall j \in [B], y_j = r_j + \mathbb{1}\{j \text{ is terminal}\} \cdot \gamma \max_{a'} Q(s'_j, a'; \theta_\tau^{\text{target}})$
            Update $\theta_{e,t+1}$ using Q-objective $\frac{1}{B} \sum_{j=1}^B (y_j - Q(s_j, a_j; \theta))^2 \big|_{\theta = \theta_{e,t}}$.
        **end for**
        Update the parameter for next episode $\theta_{e+1,1} \leftarrow \theta_{e,H+1}$.
    **end for**
    Update the target parameter $\theta_{\tau+1}^{\text{target}} \leftarrow \theta_{M,H+1}$.
**end for**

---

### A.1.3 TRAINING HARDWARE

For all experiments, we used a local server running Ubuntu 20.04 LTS, with 2 Intel Xeon Silver 4214R CPUs (2.40 GHz) and 196GB of RAM. No GPUs were used in our experiments.

### A.1.4 RELEVANT CODE

The following files are included as part of the zipped supplementary materials. We describe the use for each file:

- `captive`: codes for the game server;
- `rules_files`: rule and board configurations;
- `rl_codes`: DQN implementation:
  - `dqn.py`: the DQN algorithm;
  - `driver_dqn.py`: experiment setup;
  - `rule_game_engine.py` and `rule_game_env.py`: codes to interact with the game server;
  - `rule_sets.py`: featurization;
  - `run_code.sh`: set the path to code of captive game server ;
  - `params/last_summer_rule/*`: experiment configuration for each rule;
  - `run_script/run_last_summer_dqn_rand.sh`: script to run the experiment.

### A.1.5 TRAINING PROCEDURE

The code can be run by changing the directory to `rl_codes` and using the following command: `bash run_script/run_last_summer_dqn_rand.sh`.

