# OpenReview forum: "The Game of Hidden Rules: A New Challenge for Machine Learning"
_ICLR.cc/2023/Conference — Submitted to ICLR 2023_

### Official Review · Reviewer_FEUd · 2022-10-20

**Confidence:** 3
**Clarity, Quality, Novelty And Reproducibility:** This paper is clear, novel and can be…
**Correctness:** 3
**Technical Novelty And Significance:** 3
**Empirical Novelty And Significance:** 3
**Recommendation:** 6

**Strength And Weaknesses:**

Strengths:

The proposed environment is novel and seems to be used to study the impact of task characteristics on learning since it is constructed specifically for configurability.

The GOHR has expressive rule syntax so that researchers can control over tasks and make precise changes to rules of interest to study how they affect algorithm performance.

Weaknesses:

This paper claims that researchers can control and design tasks in GOHR. In my understanding, this controllable environment should have the ability to compare different tasks once the task is designed so that researchers can change the task design in advance. However, the examples in this paper utilize the cumulated error curves to evaluate the difficulty between tasks. If the evaluation is realized after the experiment, what is the use of this so-called controllable task since we need to carry out experiments after designing tasks to compare different tasks.



**Summary Of The Paper:**

This paper introduce a novel learning environment to study how task characteristics affect measured difficulty for the learner. Besides, it detailed the expressive rule syntax in the environment and showed how to control over tasks. Some example results and analysis are presented in the paper and additional benchmark rules are provided in public site.

**Summary Of The Review:**

Novel environment but I'm not sure about the usefulness.

---

### Official Review · Reviewer_Ccmc · 2022-10-25

**Confidence:** 3
**Correctness:** 4
**Technical Novelty And Significance:** 3
**Empirical Novelty And Significance:** 2
**Recommendation:** 8

**Clarity, Quality, Novelty And Reproducibility:**


The paper is well-written and clear.  There are no reproducibility issues.

**Strength And Weaknesses:**

Strengths:

-- Paper is clear and well written.

-- A platform and a suite of tools is provided so researchers can do
   different analyses (the authors show some of what one can do).

Weaknesses (More details after the list):

-- Because the setting is so contrived (by definition or from the
 get-go!) and combinatoric, how well any findings translate to
 real-world performance or difficulty is always questionable (this is
 admittedly an issue with any game and/or synthetic setting). If the
 authors could provide an example of a real-world problem (drug
 discovery, optimization, planning, etc) where engaging in this study
 could provide insights, that would strengthen the paper.

-- [related to above] It would be good to have more uncertainty (see
 below). The authors focus on and motivate determinism, perhaps to
 allow for systematic study and comparisons (page 2, deterministic
 rules leading to 'clear distinctions.. [among].. learning tasks', but
 uncertain rules and environments could be insightful too and lead to
 clear distinctions..).


----------------------------------

More details:


-- How about non-deterministic rules and incomplete rule sets? so that
 the agent needs to learn or anticipate that the world is noisy and
 uncertain? This would be more applicable to real life: (often) no
 rule learned is perfect, and no rule set is complete!  As the authors
 explained, there could be some non-determinism in how rules are
 applied (eg in the order the rules become applicable), but that's
 very limited.

-- For the above case, perhaps the agent needs to learn when to stop
 learning, ie it has learned all that it could and can do no better
 (which changes the interface to the platform).

-- More non-stationarity could be another complicating factor: the rule
 set changing over time, but not in a deterministic/periodic fashion.



**Summary Of The Paper:**

The authors describe an expressive platform, GOHR, for generating game
rules (of a certain number and complexity) and for evaluating how fast
the various agents figure out the rules.  Such a platform can lead to
insights into difficulty of rule sets, and relative capabilities of
different learning approaches, through systematic studies. For
instance, are there strategies that are uniformly good, or in some
sense pareto optimal? The authors also present an example MDP-based RL
agent, and a few different types of rules, and study the relative
hardness of learning those rules with respect to the agent (how long
the agent takes, until it makes no mistakes).


**Summary Of The Review:**


The work in trying to give researchers control over level difficulty in a space of tasks, and providing an easy to use
platform is a good contribution.  I am concerned the space remains too narrow and may not generalize to the real world.
However, overall, I am positive on the paper.

---

### Official Review · Reviewer_9f8z · 2022-10-26

**Confidence:** 4
**Correctness:** 3
**Technical Novelty And Significance:** 2
**Empirical Novelty And Significance:** 2
**Recommendation:** 3

**Clarity, Quality, Novelty And Reproducibility:**

Paper is pretty clear. It is somewhat novel, in the sense that I have not seen this game before, but I have seen plenty of games where the transition dynamics can change from episode to episode. It seems fairly easily reproducible, particularly since the paper is not focused on any specific empirical results, and the game proposed is very flexible.

**Strength And Weaknesses:**

Strengths
-----------
The authors propose an interesting game idea, certainly, and I can certainly see it could be useful as a domain.

Weaknesses
------------
It's not clear to me whether proposing this game is in itself enough of a contribution to merit publication in a venue such as ICLR. Especially since there is not a sufficiently strong analysis of how different learning algorithms fair in this domain, at least as baseline, which is something I would expect given the nature of this paper.

Furthermore, it's not clear to me what uniquely qualifies this game to study the way task parameters impact learning, more than countless other domains used in contemporary ML, from Starcraft and FLOW to classic games such as chess and Go, which all have their configurable characteristics, which researchers indeed tweak. Furthermore there are plenty of games in which the transition dynamics may change from episode to episode - what makes this domain better suited for study than any of the countless games authors have come up with in decades of ML research? I am not claiming this is not the case, just that I don't think it's sufficiently clear from the paper what the differentiators are.

**Summary Of The Paper:**

In this paper the authors present a new test domain for ML ideas, called the Game of Hidden Rules, meant to help researchers disentangle how different aspects of the task and setting impact the level of difficulty for learning algorithms. In each episode of the game, the learning agent is presented with a game board containing game pieces, each drawn from a configurable set of shapes and colors. The objective is to remove game pieces from play by dragging and dropping them into buckets located at each corner of the game board. A hidden rule determines which pieces may be placed into which buckets at a given point in the game play. For instance, a rule might assign game pieces to specific buckets based on their shape or color. If the agent attempts a move that is not permitted, the piece remains unmoved. Rules can also be nonstationary, for instance by changing the logic of which moves are allowed during play.

**Summary Of The Review:**

This is an interesting idea but I don't think there's enough of a contribution here at the moment to be useful to the general ICLR community. There is a ton of potential though, so I truly hope the authors bulk up this study with more results and insights before calling upon us to embrace it as a new central test domain for ML research.

---

### Decision · Program_Chairs · 2023-01-20

**Decision:**

Reject

**Justification For Why Not Higher Score:**

The paper proposes an interesting new benchmark domain, but fails to systematically show it can be used to derive novel insights. As a result, the contribution is not sizable enough to warrant acceptance,

**Justification For Why Not Lower Score:**

N/A

**Metareview: Summary, Strengths And Weaknesses:**

The paper proposes a novel learning environment, the Game of Hidden Rules (GOHR), which aims to provide the tools needed to understand task characteristics and their effects on task difficulty (e.g., for a given reinforcement learning algorithm). The GOHR is a board clearing game, and allows researchers to specify boards and rules for clearing them. The paper details the learning platform and gives and example algorithm with performance evaluation.

Initial reviews of this paper noted that the proposed learning environment constitutes an interesting and novel problem domain. The platform gives a lot of control to researchers, and enables a range of analyses. The paper was also found to be clear and well-written.

Several questions and concerns were raised. Questions included: how this specific set of configurable characteristics improves over existing benchmarks and learning environments, how a researcher would go about designing task distributions without substantial experimentation to determine task difficulty, and how results and insights might generalize to other domains, including real-world applications. One reviewer encouraged moving towards real-world aspects by introducing additional capacity for stochasticity or non-determinism. A major concern raised was whether simply proposing the benchmark was a sufficient contribution. The reviewer suggested that additional analysis was needed to support claims, and show how the proposed learning platform could be used to develop interesting novel insights.

Some of the concerns that were initially raised by the reviewers were addressed by the authors through clarifications and discussions throughout the rebuttal phase.

At the same time important concerns remain. First, the interplay between task design and evaluation of task difficulty has not been fully addressed. Second, the relationship between the relatively contrived task here and potential real world applications has not been clarified fully. Third, the authors did not convincingly demonstrate that the proposed learning environment complements existing domains in that it enables valuable novel insights about learning approaches. Ideally, this would be done empirically, e.g., with a compelling case study of how the platform is used to drive insight.

Given the remaining concerns, the recommendation is not to accept at this stage.